# Iterative Feature Matching: Toward Provable Domain Generalization with Logarithmic Environments

**Yining Chen**
Stanford University
cynnjjs@stanford.edu

**Elan Rosenfeld**
Carnegie Mellon University
elan@cmu.edu

**Mark Sellke**
Stanford University
msellke@stanford.edu

**Tengyu Ma**
Stanford University
tengyuma@stanford.edu

**Andrej Risteski**
Carnegie Mellon University
aristesk@andrew.cmu.edu

## Abstract

Domain generalization aims at performing well on unseen test environments with data from a limited number of training environments. Despite a proliferation of proposed algorithms for this task, assessing their performance both theoretically and empirically is still very challenging. Distributional matching algorithms such as (Conditional) Domain Adversarial Networks [12, 28] are popular and enjoy empirical success, but they lack formal guarantees. Other approaches such as Invariant Risk Minimization (IRM) require a prohibitively large number of training environments—linear in the dimension of the spurious feature space $d_s$—even on simple data models like the one proposed by Rosenfeld et al. [37]. Under a variant of this model, we show that ERM and IRM can fail to find the optimal invariant predictor with $o(d_s)$ environments. We then present an iterative feature matching algorithm that is guaranteed with high probability to find the optimal invariant predictor after seeing only $O(\log d_s)$ environments. Our results provide the first theoretical justification for distribution-matching algorithms widely used in practice under a concrete nontrivial data model.

## 1 Introduction

Domain generalization aims at performing well on unseen environments using labeled data from a limited number of training environments [8]. In contrast to transfer learning or domain adaptation, domain generalization assumes that neither labeled nor unlabeled data from the test environments is available at training time. In the empirical literature, invariance of a "signal" feature distribution conditioned on the label, i.e. $P(\Phi(x) \mid y)$, is the underlying assumption in widely adopted algorithms such as Correlation Alignment (CORAL) [43, 42], Maximum Mean Discrepancy (MMD) [13, 25], and (Conditional) Domain Adversarial Networks [12, 28]. Distribution-matching algorithms regularize the distribution of intermediate representations of examples from different environments to be closer in some metric. They are among the top-performing methods for domain generalization benchmarks like PACS [24], VLCS [11], and DomainNet [34], but they lack formal guarantees. Previous empirical works usually characterize the performance of these algorithms using generalization bounds based on divergence between domain distributions [7, 31], but those bounds are vacuous for the typical benchmarks for domain generalization and thus cannot explain their success. For example, when different environments have disjoint supports (as is the case in the above-mentioned benchmarks) the $\mathcal{H}$-divergence [7] is 1 even for the linear hypothesis class. To obtain useful guarantees, it is necessary to study data models that encode structure reflective of settings of interest. Prior works attempting to theoretically characterize the performance of feature matching algorithms emphasize

36th Conference on Neural Information Processing Systems (NeurIPS 2022).

lower bounds [49, 44]. In this work, we seek to give the first *positive* theoretical justification for feature matching algorithms.

Another common assumption in the literature is invariance of the label distribution conditioned on the signal features. Invariant Risk Minimization (IRM) [3] assumes $\mathbb{E}[y \mid \Phi(x)]$ is invariant, and follow-up works assume invariance of higher moments [46, 18, 30, 6]. However, empirical results for these algorithms are mixed: Gulrajani and Lopez-Paz [15], Aubin et al. [4] present experimental evidence that these methods do not consistently outperform ERM for either realistic or simple linear data models, when fairly evaluated.

Recent theoretical works [37, 19] also question the theoretical foundations of IRM and its variants, shedding light on their failure conditions. These works study specific data generative models; a common assumption is that, conditioned on the label, some invariant features have identical distribution for all environments, and other spurious features have varying distributions across environments. The goal of these methods is then to obtain an *invariant predictor*, i.e. a classifier which uses only the invariant features. These works also often assume each training environment contains infinite samples. Thus, **the central measure of domain generalization is the number of environments needed to recover an invariant predictor—we refer to this measure as the *environment complexity* of a learning algorithm.** Rosenfeld et al. [37] show even for a simple generative model and linear classifiers, the environment complexity of IRM—and other objectives based on the same principle of invariance—is at least as large as the dimension of the spurious latent features, $d_s$. Further results by Kamath et al. [19], Ahuja et al. [2] also point to a linear environment complexity. Although the models in these works are simple, they help elucidate why existing algorithms fail and can help inform better algorithmic design.

IRM's linear environment complexity is prohibitive for realistic applications. Domain generalization benchmarks have fewer training environments than spurious features dimensions—PACS, VLCS, DomainNet have 4-6 total environments but the stylistic variations between domains are likely high-dimensional. In this paper, we show that a variant of distributional matching algorithm (Algorithm 1) finds the optimal invariant predictor with sublinear (even logarithmic) environment complexity.

## 1.1   Our contributions

Conceptually, we propose a "smoothed covariance" extension of the data model in Rosenfeld et al. [37], which is rich enough to establish an environment complexity separation between IRM/ERM and feature matching algorithms. More precisely, we show that ERM and IRM can fail to find the optimal invariant predictor with fewer than $d_s + 1$ training environments (Theorems 4.2, 4.3), while a relatively simple and natural algorithm based on iterative feature matching (IFM) finds the optimal invariant predictor with $O(\log d_s)$ environments.

Moreover, our analysis highlights the value of iterative matching, much similar to matching at multiple layers in a deep neural network. IFM iteratively projects the features to a lower dimension while matching the label-conditioned feature distributions on a small, disjoint subset of the training environments. A projection inducing invariance in the non-invariant features across one subset of environments is unlikely to do so for a different subset, so each projection only removes (a constant fraction of) spurious feature dimensions with high probability. Intuitively, the iterative scheme prevents the different environments from "colluding" to create a solution which depends on spurious features. As a result, IFM recovers the optimal invariant predictor after $O(\log d_s)$ rounds and uses $O(1)$ environments per round, thus requiring $O(\log d_s)$ environments.

Our techniques for proving the upper and lower bounds may be of independent interest. The upper bound (Theorem 5.1) is proved by showing that any projection matrix that uses many spurious dimensions cannot match feature covariances in a large set of training environments. This is done via intricate matrix concentration bounds and decoupling inequalities [10].

We derive the exact lower bound for IRM (Theorem 4.3) using tools from differential topology. Each environment provides an ellipsoidal constraint on the solution, and we prove that there exists a non-trivial intersection of these constraints (besides the origin, which corresponds to the "intended" solution). Our key lemma shows that the total number of intersections between two manifolds of complementary dimensions $k, d - k$ is even when certain tranversality conditions hold, implying that the origin cannot be the only solution.

Finally, to corroborate the advantages of the proposed algorithm, we perform experiments on a Gaussian dataset and a semi-synthetic Noised MNIST dataset [23], where the background noise spuriously correlates with the label. Our results in Section 6 suggest that practitioners may benefit from feature matching algorithms when the distinguishing property of the signal feature is indeed conditional distributional invariance, and may get additional advantage via matching at multiple layers with diminishing dimensions, echoing existing empirical observations [27, 29].

## 2 Preliminaries

### 2.1 Domain generalization

In domain generalization, we are given a set of $E$ training environments $\mathcal{E}_{tr}$ indexed by $e \in [E]$,[1] and a set of test environments $\mathcal{E}_{ts}$. For environment $e$ we have $n$ examples $\{(X_i^e, Y_i^e)\}_{i=1}^n$ drawn from the distribution $P_e$. In this work we study the infinite sample limit $n \to \infty$ so as to separate the effect of limited training environments from that of limited samples *per* environment, as is done in previous theoretical works [37, 19]. Let $\mathcal{X}, \mathcal{P}, \mathcal{Y}$ denote the space of inputs, intermediate features, and labels. For a featurizer $\Phi : \mathcal{X} \to \mathcal{P}$ and classifier $w : \mathcal{P} \to \mathcal{Y}$, their risk on environment $e$ is denoted by $R_{\Phi,w}^e = \mathbb{E}_{(X,Y) \sim P_e}[l(w \circ \Phi(X), Y)]$ for any common loss function $l$. In this paper we focus on $\mathcal{Y} = \{\pm 1\}$, linear featurizers $\Phi(X) = UX$ for $U \in \mathbb{R}^{k \times d}$, and unit-norm predictors $\widehat{Y} = \text{sgn}\,(w^\top U x)$ where $w \in \mathbb{R}^k$ and $\|w^\top U\|_2 = 1$ for some feature dimension $k \leq d$ chosen by the algorithm. A predictor's 0-1 risk on environment $e$ is denoted by $R_{U,w}^e = \Pr_{(X,Y) \sim P_e}[\text{sgn}\,(w^\top U X) \neq Y]$. We focus on unit-norm predictors because we evaluate on the 0-1 risk on test environments, which are invariant to the scaling of $w^\top U$ under our data model.

### 2.2 Baseline algorithms

We analyze the performance of our proposed method and compare it to two baseline algorithms, ERM and IRM. ERM learns a classifier that minimizes the average loss over all training environments, where $l$ is any common training loss such as the logistic loss:

$$\min_{w \in \mathbb{S}^{d-1}} \frac{1}{E} \sum_{e \in [E]} \mathbb{E}_{(X,Y) \sim P_e}[l(w^\top X, Y)].$$

IRM learns a featurizer $\Phi(X) \in \mathbb{R}^k$ such that the optimal classifier on top of the featurizer is invariant across training environments. As we focus on linear classifier, it is equivalent to learning a linear transformation $U \in \mathbb{R}^{k \times d}$ such that it induces a classifier $w$ that is optimal for all $e \in \mathbb{E}_{tr}$:

$$\min_{U \in \mathbb{R}^{k \times d}, w \in \mathbb{R}^k, \|w^\top U\|_2 = 1} \frac{1}{E} \sum_{e \in [E]} \mathbb{E}_{(X,Y) \sim P_e} l((w^\top U X), Y)$$

$$\text{s.t.} \quad w \in \arg\min_{w' \in \mathbb{R}^k} \mathbb{E}_{(X,Y) \sim P_e}[l((w'^\top U X), Y)], \forall e \in \mathcal{E}_{tr}.$$

This is objective is *not* the same as feature distribution matching; IRM only tries to match the first moment. Observe that this constrained objective is intended to solve a minimax domain generalization problem, as opposed to ERM which is typically viewed as minimizing the risk in expectation.

## 3 Problem setup

We first recall the data model from Rosenfeld et al. [37]. We assume without loss of generality that the label $Y$ is uniformly randomly drawn from $\{\pm 1\}$ (extension of our theorems to $Y = 1$ with probability $\eta \neq 0.5$ is straightforward). Latent variable $Z$ consists of invariant features $Z_1 \in \mathbb{R}^r$ and spurious features $Z_2 \in \mathbb{R}^{d_s}$ where $d_s = d - r$. The number of spurious features can be much larger than the number of invariant features, i.e. $d_s \gg r$. The input $X \in \mathbb{R}^d$ is generated via a linear

---

[1]We define $[n] = \{1, \ldots, n\}$; $\mathbf{0}_{n \times m} \in \mathbb{R}^{n \times m}$ denotes an all-zero matrix; $\mathbb{S}^d$ is the unit sphere in $\mathbb{R}^{d+1}$; $\text{sgn}\,(c) \in \{\pm 1, 0\}$ is the sign of scalar $c \in \mathbb{R}$. † denotes the Moore-Penrose pseudo-inverse.

transformation of latent variable $Z$, i.e. $X = SZ$ for a matrix $S \in \mathbb{R}^{d \times d}$ such that its left $r$ columns have rank $r$ (so that there are $r$ invariant dimensions).

For each training environment indexed by $e \in [E]$, the invariant features conditioned on $Y$ are drawn from a Gaussian distribution with mean $Y \cdot \mu_1 \in \mathbb{R}^r$ and nonsingular covariance $\Sigma_1 \in \mathbb{R}^{r \times r}$. The spurious features conditioned on $Y$ have mean $Y \cdot \mu_2^e \in \mathbb{R}^{d_e}$ and covariance $\Sigma_2^e \in \mathbb{R}^{d_e \times d_e}$ where $\mu_e$'s and $\Sigma_e$'s vary across $e \in [E]$. The assumption of symmetric class center with respect to the origin can also be relaxed. Define $\mu^e = [\mu_1, \mu_2^e]$ and $\Sigma^e = [\Sigma_1, \mathbf{0}_{r \times d_s}; \mathbf{0}_{d_s \times r}, \Sigma_2^e]$. The overall data model for training environments is summarized below:

$$Y \overset{iid}{\sim} \text{unif}\{\pm 1\}$$
$$Z_1 | Y \sim N(Y \cdot \mu_1, \Sigma_1) \in \mathbb{R}^r$$
$$Z_2 | Y \sim N(Y \cdot \mu_2^e, \Sigma_2^e) \in \mathbb{R}^{d_s}$$
$$Z = [Z_1, Z_2] \in \mathbb{R}^d$$
$$X = SZ.$$

Since the goal of invariant feature learning is to learn a predictor that only uses the invariant features, one reasonable measure for domain generalization is a predictor's performance on test environments where the spurious features $Z_2$ are drawn from a different distribution—in particular, they are usually chosen adversarially. A classifier that predicts using the spurious features will perform badly on such test environments. When modeling the test environments, we consider the difficult scenario where there is one corresponding test environment for each training environment, whose parameters are the same except that the spurious means are flipped. Formally, for each environment $e \in \mathcal{E}_{tr}$ we construct a corresponding test environment $e' \in \mathcal{E}_{test}$ where

$$Z_2 \sim N(-Y \cdot \mu_2^e, \Sigma_2^e) \in \mathbb{R}^{d_s}.$$

In this setting where the observations $X$ are a linear function of the latents $Z$, Rosenfeld et al. [37] assume that the covariances of spurious features are isotropic and vary only in magnitude:

**Assumption 3.1** (Data model for spurious covariances in Rosenfeld et al. [37]).

$$\Sigma_2^e = \sigma_e^2 I_{d_s}, \text{ where } \sigma_e \text{ is a scalar for an environment indexed by } e.$$

We consider a generalized model where the covariances of spurious features for each environment is a generic random PSD matrix, instead of only random in scaling:

**Assumption 3.2** (Data model for spurious covariances in this work).

$$\Sigma_2^e \sim \overline{\Sigma_2^e} + G_e G_e^\top, \text{ where } \overline{\Sigma_2^e} \in \mathbb{R}^{d_s \times d_s} \text{ is arbitrary (and can be adversarial), and}$$
$$[G_e]_{i,j} \overset{iid}{\sim} N(0,1) \text{ for all } i,j \in [d_s]. \text{ Furthermore,} \max_e \|\overline{\Sigma_2^e}\|_2^2 \leq D.$$

Assumption 3.2 allows the covariance matrix of the spurious features to be *almost worst-case*: it is an arbitrary matrix, plus a Gaussian perturbation. This is a common assumption in algorithmic complexity called *smoothed analysis* [41], as we allow the parameters to be a random smoothing of arbitrary parameters.

While our data model is simple, it has already been used as a sandbox to understand algorithms like IRM [37, 19] because it captures some important aspects of real-life data like latent variables and correlations between the labels and the spurious features.

In the next section, we show that baseline algorithms ERM and IRM still have $\Omega(d_s)$ environment complexity under assumption 3.2, whereas our iterative feature matching algorithm (Algorithm 1) requires only $O(\log d_s)$ training environments. Note that the environment complexity of our algorithm only depends logarithmically on the norm bound $D$.

## 4 Main results

Armed with assumption 3.2, we now present our main results. We begin by presenting our algorithm based on iterative feature matching. We then provide formal guarantees for its environment complexity in comparison to ERM and IRM.

**Algorithm 1** Iterative Feature Matching (IFM) algorithm

---

**Require:** Invariant feature dimension $r$, total feature dimension $d$, number of training environments $E = |\mathcal{E}_{tr}|$, infinite samples $\{(X_i^e, Y_i^e)\}_{i=1}^{\infty} \sim P_e$ from each environment $e \in \mathcal{E}_{tr}$, an array of integers indicating the number of training environments to use at each iteration $E_1, E_2, \ldots$.

1: $r_0 \leftarrow d, t \leftarrow 0$.
2: **while** $r_t > r$ **do**
3:      $t \leftarrow t + 1$.
4:      Uniformly randomly sample $E_t$ training environments without replacement.
5:      Binary search between 1 and $r_{t-1}$ to find the maximum dimension $r_t$ such that there exists orthonormal $U_t \in \mathbb{R}^{r_t \times r_{t-1}}$ and $C_t \in \mathbb{R}^{r_t \times r_t}$, where for all $e \in \mathcal{E}_t$,

$$\mathbb{E}_{(X,Y) \sim P_e}[U_t \ldots U_1 X X^\top U_1^\top \ldots U_t^\top | Y] = C_t. \tag{4.1}$$

6: Return a classifier on projected features that minimizes the average risk

$$\widehat{w} = \min_{w \in \mathbb{S}^{r-1}} \frac{1}{E} \sum_{e \in [E]} \mathbb{E}_{(X,Y) \sim P_e} l(w^\top U_t \ldots U_1 X, Y).$$

---

## 4.1 Iterative feature matching algorithm

We hope to recover the invariant features by imposing constraints which are satisfied by only those features (i.e., they are not satisfied by the spurious features). A natural idea is to match the feature means and covariances across $\mathcal{E}_{tr}$. Since $\mu_1, \Sigma_1$ are constant, any orthonormal featurizer $U \in \mathbb{R}^{r \times d}$ such that $US$ has only non-zero entries in the first $r$ rows yields invariant means $US\mu^e$ and covariances $US\Sigma^e S^\top U^\top$. Thus we need $E$ large enough such that any $U' \in \mathbb{R}^{r \times d}$ using spurious dimensions cannot match the means and covariances. Informally, for each $e \in [E]$ we get $r \times r$ equations from matching covariances $U\Sigma^e U^\top = C$, and we have $r \times d$ parameters to estimate in $U$. Rough parameter counting suggests that if we match covariances of all $E$ environments jointly, we need at least $E > d/r$ environments to find a unique solution. Our key observation is that, due to the independence of randomness in $\Sigma_2^e$, we can split $E$ environments into disjoint groups $\mathcal{E}_1, \ldots, \mathcal{E}_T$, and use $\mathcal{E}_t$ to train an orthonormal featurizer that shrinks the feature dimensions from $r_{t-1}$ to $r_t$. Thus, in each round we shrink the dimension by a constant factor using a constant number of environments. The main theoretical challenge that remains is to show that in each iteration, with high probability, *only* spurious features are projected out.

This brings us to IFM (Algorithm 1), which proceeds in $T = O(\log d_s)$ rounds. Starting with an input dimension $r_0 = d$, each round we learn an orthonormal matrix $U_t$ projecting features from $r_{t-1}$ to $r_t$ dimensions so that the feature covariances after projection match across a fresh set of training environments. In practice, for each choice of $r_t$, starting from random initialization, we perform SGD on objective $\min_{U_t \in \mathbb{R}^{r_t \times r_{t-1}}} \sum_{e,e' \in \mathcal{E}_t} \|U_t(\Sigma^e - \Sigma^{e'})U_t^T\|_F^2 + \lambda \|U_t^\top U_t - I\|_F^2$ until the objective is $\epsilon$-close to 0. To ensure that all invariant dimensions are preserved, we always find the projection with the maximum possible dimension $r_t$ that still matches the covariances (i.e. admits an $\epsilon$-optimal solution $U_t$), until we are left with only $r$ dimensions.

Among algorithms used in practice, IFM is most similar to CORAL [43]. We study IFM since it is more amenable to theoretical analysis. The differences between IFM and CORAL are: first, CORAL does not enforce that the featurizer is orthonormal; second, IFM learns to extract features in an unsupervised manner, whereas CORAL jointly minimizes the supervised loss and feature distribution discrepancy; third, IFM matches the feature distributions at multiple layers and uses a disjoint set of environments for each layer—this iterative process is necessary for the theoretical guarantees we provide—but CORAL matches only at the last layer. Despite these differences, our theoretical results serve as a justification for using feature matching algorithms in general, when the distinguishing attribute of signal vs. spurious features is that the former have invariant distributions across all environments. In section 6 we empirically show that adding the core features of IFM to CORAL can improve test accuracy.

The following theorem states that the environment complexity of IFM is logarithmic in the spurious feature dimension. A proof sketch is given in Section 5.

**Theorem 4.1** (IFM upper bound). *Under assumption 3.2, suppose at each round IFM uses $|\mathcal{E}_t| = \tilde{\Omega}(1)$[2] training environments, with probability $1 - \exp(-\Omega(d_s))$, IFM terminates in $O(\log d_s)$ rounds and outputs $\widehat{w} = w^*$.*

As a remark, we note that there is in fact a simple algorithm that achieves $O(1)$ environment complexity under Assumption 3.2 (see Appendix D). However, this algorithm is extremely brittle and reliant on very specific aspects of the data model (such as Gaussianity) and it cannot be extended to other settings, whereas feature matching algorithms are regularly applied to general model architectures and real-world datasets. The goal of this paper is to provide theoretical justification for distribution matching algorithms and to investigate why they may outperform ERM and IRM, rather than to solve a specific data model.

## 4.2 ERM and IRM still have linear environment complexity

In the previous section, we showed that IFM has low environment complexity thanks to the additional structure we imposed in our model. However, it is possible that this additional structure also allows ERM and IRM to succeed. These next two results demonstrate that this is not the case.

**ERM has low test accuracy** In contrast to IFM, ERM still suffers from linear environment complexity under Assumption 3.2. The first theorem says there are hard instances where the ERM solution has worse-than-random performance on the test environments.

**Theorem 4.2** (ERM lower bound). *Suppose $E \leq d_s$, parameters $\mu_1 \in \mathbb{R}^r$, $\Sigma_1 = \sigma_1^2 I_r$, $\mu_2^e \in \mathbb{R}^{d_s}$, $\overline{\Sigma_2^e} = \sigma_2^2 I_{d_s}$ (recall $\Sigma_2^e \sim \overline{\Sigma_2^e} + G_e G_e^\top$). Then any unit-norm linear classifier which achieves accuracy $\geq \Phi\left(\frac{2\|\mu_1\|}{\min(\sigma_1,\sigma_2)}\right)$ on all training environments will suffer 0-1 error at least $\frac{1}{2}$ on every test environment with flipped spurious mean, where $\Phi$ is the standard Normal CDF.*

A complete proof of Theorem 4.2 is in Appendix B.3. Note that it is quite reasonable to assume that the ERM solution satisfies the accuracy condition. In particular, it is common to model the spurious features as having much greater magnitude than the invariant features, since they have much greater dimensionality. For example, with a unit-norm mean we would expect $\|\mu_1\|^2 \approx r/d$, $\|\mu_2^e\|^2 \approx d_s/d$. Then for $r \ll d$ and $\sigma_1, \sigma_2 = \omega(1/\sqrt{d})$ one can verify that $2\|\mu_1\|/\min(\sigma_1,\sigma_2) = o(\sqrt{r/d})$ is very close to 0, meaning the lower bound $\Phi\left(\frac{2\|\mu_1\|}{\min(\sigma_1,\sigma_2)}\right)$ is only slightly larger than $\frac{1}{2}$.

**IRM fails to learn invariant features** Our next theorem proves that even under Assumption 3.2, IRM is still not guaranteed to find an invariant predictor. We can show this by proving that when $E \leq d_s$, we can find a featurizer that only uses spurious dimensions, i.e., $u_s \in \mathbb{R}^{d_s}$, such that $u_s^\top \Sigma_2^e u_s = u_s^\top \mu_2^e$ for all $e \in \mathcal{E}_{tr}$. If so, the optimal predictor on top of features $u_s^\top Z_2$, $\widehat{w^e} = (u_s^\top \Sigma_2^e u_s)^{-1} u_s^\top \mu^e$ is invariant across all $e$, and can therefore be the preferred solution IRM when the spurious features have greater magnitude than the invariant features on the training environments.

**Theorem 4.3** (IRM lower bound). *Suppose $E \leq d_s$. If $\mu_2^1, \ldots, \mu_2^E \in \mathbb{R}^{d_s}$ are linearly independent, then there exists $u_s \in \mathbb{R}^{d_s}$, $\|u_s\|_2 > 0$, such that $u_s^\top \Sigma_2^e u_s = u_s^\top \mu_2^e$ for all $e \in [E]$.*

*Proof Sketch.* Observe that each environment provides an ellipsoidal constraint $E_e = \{u_s \in \mathbb{R}^{d_s} : u_s^\top \Sigma_2^e u_s - u_2^\top \mu_2^e = 0\}$. The origin is a trivial intersection. We prove the existence of a non-trivial intersection using tools from differential topology. The key lemma is that the total number of intersection points between two manifolds of complementary dimensions $k, d-k$ is even when certain tranversality conditions hold. Using these techniques, we show that $|\bigcap_e E_e| \geq 2$ for almost all matrices $\Sigma_2^1, \ldots, \Sigma_2^E$, as long as the means are linearly independent. $\qquad\square$

A complete proof of Theorem 4.3 is in Appendix B.4.

---

[2] $\tilde{\Omega}(\cdot)$ hides logarithmic factors in $D, r, d_s$.

# 5 Proof sketch for the main upper bound Theorem 4.1

To argue that IFM outputs a featurizer $U_1 \ldots U_T$ that does not use the spurious features, we need to show that the right $d_s$ columns of matrix $U_T \ldots U_1 S$ are all-zero. The main lemma below says that this happens with high probability if we match $\tilde{\Omega}(1)$ environments at every iteration,

**Lemma 5.1.** *If for all $1 \leq t \leq T$, $|\mathcal{E}_t| = E_t = \Omega\left(c(\log\left(D/d_s\right) + \log d_s)\right)$, and $U_1, \ldots, U_T$ are the orthonormal matrices returned by IFM, then with probability $1 - \exp\left(-\Omega(d_s)\right)$, $r_t - r < (r_{t-1} - r + 1)/c$ for all $t$, and if we write $U_T \ldots U_1 S = [A, B]$, where $B \in \mathbb{R}^{r \times d_s}$, then $B = \mathbf{0}_{r \times d_s}$.*

Theorem 4.1 follows from Lemma 5.1 as follows: We take $c = 2$ and the algorithm terminates in $T = O(\log d_s)$ rounds. Therefore with an environment complexity of $O(\log d_s)$, we learn a feature extractor $U = U_T \ldots U_1$ that does not use any spurious dimensions. Since $U$ is orthonormal, it must contain all signal dimensions. The predictor on top of this representation uses all and only signal dimensions, so with high probability, IFM outputs $\widehat{w} = w^*$.

The first step towards proving Lemma 5.1 is to show that with high probability, any *one-layer* featurizer $Q_1 \in \mathbb{R}^{k_1 \times d_s}$ that uses *only* spurious dimensions cannot match feature covariances from $\tilde{\Omega}(d_s/k_1)$ environments. If a featurizer $U_1 \in \mathbb{R}^{r_1 \times d}$ uses $k_1$ spurious dimensions, there is a corresponding rank-$k_1$ featurizer $Q_1 \in \mathbb{R}^{k_1 \times d_s}$ that uses *only* spurious dimensions. So Lemma 5.2 implies that any $U_1$ that matches covariances in $\mathcal{E}_1$ must use at most $d_s/E_1$ spurious dimensions. We will then apply this argument recursively until we have 0 spurious dimensions.

**Lemma 5.2** (Informal version of Lemma B.3). *For any integer $2 \leq k_1 \leq d_s/2$, when $|\mathcal{E}_1| = E_1 = \Omega\left(\frac{d_s - k_1}{k_1 - 1} \max\left\{1, \log\left(\frac{D}{(k_1 - 1)d_s}\right), \log\left(\frac{d_s}{k_1 - 1}\right)\right\}\right)$, with probability $1 - O(\exp\left(-d_s\right))$, no orthonormal $Q \in \mathbb{R}^{k_1 \times d_s}$ satisfies that for some constant $C_1 \in \mathbb{R}^{k_1 \times k_1}$.*

$$\forall e \in [E_1], \quad Q\Sigma_2^e Q^\top = C_1. \tag{5.1}$$

Section B.1 gives a proof sketch. The formal statement Lemma B.3 and its proof can be found in Appendix B.2.

The next claim says that Lemma 5.2 can be applied iteratively, i.e. fixing a featurizer from previous iterations that uses $k_{t-1}$ spurious dimensions, with high probability, any $U_t$ that matches features from $\Omega\left(k_{t-1}/k_t\right)$ new environments uses at most $k_t$ spurious dimensions.

**Corollary 5.3** (Informal version of Corollary B.7). *Suppose $2 \leq k_t \leq k_{t-1}/2 \leq d_s/2$. When $|\mathcal{E}_t| = E_t = \Omega\left(\frac{k_{t-1} - k_t}{k_t - 1} \max\left\{1, \log\left(\frac{D}{(k_t - 1)d_s}\right), \log\left(\frac{d_s}{k_t - 1}\right)\right\}\right)$, for fixed orthonormal $P \in \mathbb{R}^{k_{t-1} \times d_s}$, with probability $1 - O(\exp\left(-d_s\right))$, no orthonormal $Q \in \mathbb{R}^{k_t \times k_{t-1}}$ satisfies $\forall e \in [E_t]$, $QP\Sigma_2^e P^\top Q^\top = C_t$ for some constant $C_t \in \mathbb{R}^{r_t \times r_t}$.*

The formal statement Corollary B.7 and its proof can be found in Appendix B.2. Lemma 5.1 follows from iterative application of Corollary 5.3, as shown in Appendix B.2.

# 6 Experiments

In light of the differences between IFM and CORAL discussed in section 4.1, we test several questions inspired by our theory: (Q1) Do feature matching algorithms (IFM and CORAL) have much smaller environment complexity compared to ERM and IRM, with finite samples drawn from data models similar to our assumptions? (Q2) Can decoupling feature matching and supervised training of the classifier (IFM) improve over joint training (CORAL)? (Q3) For neural network featurizers, can matching feature distributions at multiple layers improve over matching at only the last layer (naive CORAL)? (Q4) Can matching disjoint sets of environments at each layer perform as well as matching all environments at all layers? (Q5) Is it important to shrink feature dimensions? We use two tasks to investigate those questions empirically. Appendix C contains additional details.

**Gaussian dataset** is a binary classification task that closely reflects our assumptions in section 3.

**Noised MNIST** is a 10-way semi-synthetic classification task modified from LeCun and Cortes [23]. Conditioned on the class, we add noise with identical mean but changing covariances across training environments. In test environments, the noise is uncorrelated with the label.

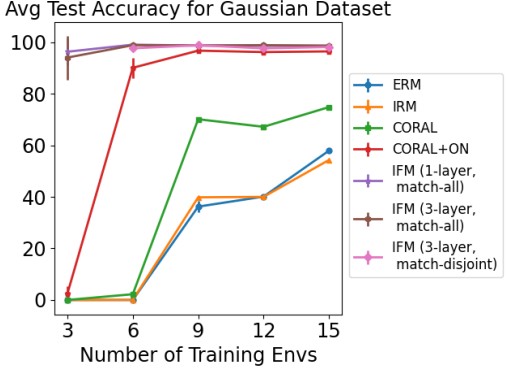

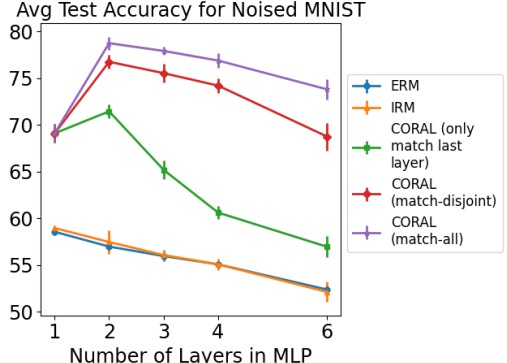

Figure 1: For Gaussian dataset, our algorithm IFM achieves highest test accuracy with the same number of training environments.

Figure 2: For Noised MNIST, matching feature distributions from multiple layers improves over naive CORAL across different architectures.

**Algorithms and architectures.** For Gaussian dataset we use linear predictors. **IRM** follows the implementation in Arjovsky et al. [3]; **CORAL** jointly minimizes average supervised loss on training environments and $L_{coral}$, which is the average of squared distances in conditonal feature means (in $l_2$ norm) and covariances (in Frobenius norm) between adjacent training environments; **CORAL+ON** adds orthonormal penalty loss $L_{on}(U) = \|UU^\top - I\|_F^2$ where $U$ is the featurizer; **IFM** is our Algorithm 1, where for each layer $U_t$, the training objective is $L_t(U_t) = \lambda_1 L_{coral} + \lambda_2 L_{on}$. We test IFM with 1 vs. 3-layer featurizers, either matching all (**match-all**) or a disjoint set of training environments (**match-disjoint**) at each layer.

For Noised MNIST we use ReLU networks with 1-6 layers. Although our theory for IFM is only for the linear setting, in practice we can extend the idea to deep network by training the classifier head together with feature matching at different layers.**CORAL (match-all)** and **CORAL (match-disjoint)** in Figure 2 are natural extensions of IFM to nonlinear models. They match at all layers post-activation, using either all (**match-all**) or a disjoint subset of training environments (**match-disjoint**) per layer. Figure 2 shows that they outperform naive **CORAL**, which matches features only at the last layer. Thus, the core ideas of IFM may be used to improve performance of existing feature matching algorithms with nonlinear models and realistic datasets (e.g. DANN, MMD).

**Results.** (Q1) Figures 1 and 2 show that IFM and CORAL have much smaller environment complexity compared to ERM and IRM in both datasets. (Q2) In Gaussian dataset, IFM improves over CORAL. (Q3) For Noised MNIST, matching feature distributions at multiple layers (CORAL match-all, CORAL match-disjoint) improves over matching at only the last layer (CORAL). (Q4) In both datasets, matching disjoint sets of environments at each layer (IFM match-disjoint, CORAL match-disjoint) is almost as good as matching all environments at all layers (IFM match-all, CORAL match-all) while saving computation. (Q5) For Noised MNIST (Table 2 in Appendix C), shrinking feature dimensions is crucial for the advantage of feature matching at multiple layers, e.g. matching features at 3 layers with widths [24, 24, 24] does not significantly improve over matching features at the last layer (CORAL). Overall, our results suggest that practitioners may benefit from feature matching algorithms when the data is similar to our assumed model, and may get additional advantage via matching at multiple layers with diminishing dimensions, echoing existing empirical works [27, 29].

## 7 Conclusion

This work presents the first domain generalization algorithm which provably recovers an invariant predictor with a number of environments that scales sub-linearly with the spurious feature dimension. Our results demonstrate that generalization which does not suffer from the "curse of dimensionality" is possible, and based on our theory we believe the use of an *iterative* approach is a key insight which could lead to additional positive results for out-of-distribution generalization. Notably, this work also represents the first theoretical justification for the empirical success of existing algorithms which use feature distribution matching. However, there remains much room for improvement: our results are

for a linear data model with fairly special assumptions. It would be interesting to analyze data models in which the observables are a non-linear function of the latent variables.

## Acknowledgements

We would like to thank Colin Wei, Sang Michael Xie, Ananya Kumar, and Ciprian Manolescu for helpful discussions. YC is supported by Stanford Graduate Fellowship. MS is supported by NSF and Stanford Graduate Fellowships. TM acknowledges support of Google Faculty Award and NSF IIS 2045685. AR and ER acknowledge support from the NSF via IIS-2211907 and the CMU/PwC DT&I Center. AR acknowledges support of an Amazon Research Award.

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
