# OpenReview forum: "Iterative Feature Matching: Toward Provable Domain Generalization with Logarithmic Environments"
_NeurIPS.cc/2022/Conference — NeurIPS 2022 Accept_

### Official Review · Reviewer_D1nV · 2022-06-21

**Rating:** 7
**Confidence:** 4
**Soundness:** 3 good
**Presentation:** 3 good
**Contribution:** 3 good

**Summary:**

This paper studies the domain generalization (DG) setting.  The authors construct a simple yet fundamental data generating procedure for DG, which shares similarities with that of _The Risks of Invariant Risk Minimization_ (henceforth RoIRM).  The authors propose an iterative algorithm which requires only logarithmically many training domains in the spurious feature dimension $d_2$ to generalize to particular test environments given infinitely many samples per domain.  This result stands in contrast to results from other works (including RoIRM)l, which show that IRM needs $O(d_s)$ environments to generalize in the linear setting.  The authors also provide several toy experiments to illustrate their results.

**Questions:**

**Test environment generation.** I won't list this as a weakness per se, but I do have questions regarding how the test environments are generated.  One natural question is whether (or to what extent) this model of test environment generation is necessary for achieving OOD generalization in log(d_s) training environments.  Let's say that I was to select completely new means $\mu^e_2$ and covariances $\Sigma_2^e$; then let

$$Z_2 \sim\mathcal{N}(Y \cdot \mu_2^e , \Sigma^e_2).$$

Would there be any hope of generalizing in this setting?  I think that this question is important because in practice, we would expect to receive data from *new domains* with new underlying parameters.  This is not to say that the setting analyzed here is less relevant; on the contrary, it seems evident that there needs to be some sort of connection between training and test domains to achieve OOD generalization.

**Dimension of style.** The authors say that

> "stylistic variations between domains are likely high-dimensional."

Is there any way to justify this?  In the case of the standard ColoredMNIST benchmark, this seems to be somewhat untrue given that the background color correlation could be controlled by a very low-dimensional parameter.  Furthermore, given the rise of large generative models which can capture semantic changes in low-dimensional feature spaces, I would tend to push back on this statement ever so slightly.

**Tranversality.**  It would be helpful if the authors could explain a bit more what they mean by this _tranversality_ condition.  I anticipate that the definition is technical, but even some intuition for this condition would be helpful.

**I.I.D.-ness.** I assume that the labels are being drawn i.i.d. from a Bernoulli, as are the latents.  This should perhaps be stated more explicitly for completeness.

**Why does $r_t$ shrink to $r$?**  It would be great if the authors could help me to understand why $r_t$ will eventually shrink to $r$.  What is stopping it from failing to shrink?

**Uniqueness.** One small question I have WRT Thm. 1 is whether $w^\star$ is necessarily unique.

**"Simple" algorithm.**  What is the basis for including this so-called "simple" alternative algorithm in the main text?  If it is mentioned, I would expect the authors to compare with it in the experiments section.  But the inclusion of this algorithm seems to be a bit orthogonal to the contribution of the paper.

**Class imbalance.**  If $\eta\neq 1/2$, how would Thm. 4.2 change?  Would the condition that the test risk under the 0-1 loss be larger than $1/2$ become $\eta$ (or perhaps $(1-\eta)$?

**ERM results.** In Thm. 4.2, I thought that we were operating under the assumption that $\bar{\Sigma^e_2}$ could be _adversarially_ chosen, and that it would not be diagonal.  It seems that one would want the assumptions of the three theorems to align; so unless I am misunderstanding something, this discrepancy calls into question whether all three results can hold under a common assumption.


**Limitations:**

The discussion in Appendix D is sufficient in my eyes.

**Strengths And Weaknesses:**

### Strengths

---

**Problem setting.**  I liked quite a bit about this paper.  I felt that first and foremost, it attempts to theoretically study what makes domain generalization hard, and in what settings we expect to generalize OOD.  This is essential, as it is not well understood in the literature when domain generalization is even possible.  The data generating procedure discussed in Section 3 seems quite natural, and it certainly extends the framework of RoIRM to the setting of (more) general covariance matrices.  The other innovation over RoIRM here seems to be the generation of the test environments.

**Definition of environmental complexity.**  This seems like quite a natural definition of complexity in DG; certainly removing the caveat that in practice we get a finite sample from each environment helps to illuminate the structure of the problem.  It would be interesting to see future works study this definition for other algorithms/data generating models.

**Theoretical work.** Thm. 1 seems to be a strong result here, because it proves the existence of an algorithm which generalizes with $\log(d_s)$ in a particular linear DG setting.  To my knowledge, such a result was not known in the literature before this work.  I believe that the proof techniques may be of independent interest here, which adds to the contribution of this paper.

### Weaknesses

---

**Over-simplification of the field of DG in the intro.**  One minor point: The introduction reads as if the *standard* assumption to make in DG is that the distributions of labels given features should be environment dependent (lines 21-22).  Furthermore, there is a claim that algorithms that rely on these assumptions "dominate" DG benchmarks (lines 23-24).  I think that both of these claims are in doubt.  There are many other works that make different assumptions on the set of stable or invariant features (see e.g. [Zhang et al., 2021] and [Robey et al., 2021], which assume domain/measurement shift).  I think this points to the fact that this paper could benefit from a related work section, which is notably absent in this submission.  WRT the point about feature-matching algorithms "dominating" the leaderboards, this seems untrue based on recent surveys: [Gulrajani & Lopez-Paz, 2020] and [Koh & Sagawa et al., 2021].  Both of these works show that ERM is a rather strong baseline relative to all other algorithms, especially relative to feature-matching algorithms.

**The term "feature matching."**  One point of confusion throughout the work was referring to "matching."  I think it would help if the authors could formally define what it means to "match" features.  The reason I bring this up is that it gets rather confusing in the non-linear case how features are being "matched."  The paper seems to assume that the reader is a priori going to know what this means, but I imagine that many readers may become confused at this point.

**Algorithm description.**  The description of the algorithm that is actually run in practice is confusing, and crucial details are missing.  One question is: How are the subsets selected?  Is $T$ a hyperparameter?  If so, how is it tuned?  Furthermore, which Alg. 1 shows us the basic steps of the algorithms, it's not clear how one would actually go about implementing this method.  In particular, the step in line 5 seems challenging.  In the end, the authors appear to tweak the implementation of CORAL by removing the ERM term.  I think it would be reasonable to expect a more robust discussion of the implementation of these methods -- which also seems to be missing from the appendices.  I am listing this as a weakness because if a contribution of this paper is to propose this algorithm as a new method, which seems to be the case based on this:

> "This work presents the first domain generalization algorithm which provably recovers an invariant predictor with a number of environments that scales sub-linearly with the spurious feature dimension."

then the paper should be responsible for connecting the pseudocode in Alg. 1 to an implementable optimization problem.

**Why doesn't CORAL work?** Related to the previous point, after reading the paper I am still unsure of why CORAL fails in the experiments whereas IFM succeeds.  Is it the fact that IFM optimizes for each $U_t$ separately in rounds, or is it the dropping of the ERM term that helps?  Perhaps this is already answered somewhere, but it was not obvious to me after reading.

**Experiments.** The experiments feel like a weak point of this paper, with the caveat that this paper seems to be more of a theoretical paper than an experimental one.  Even still, I think it wouldn't be unreasonable to expect a stronger set of experiments here.  Some comments:

* The Noisy MNIST dataset doesn't seem to satisfy the assumption concerning $Z_2$ below line 136.
* The authors claim that

> "Our results in Section 7 suggest that practitioners may benefit from feature matching algorithms when the distinguishing property of the signal feature is indeed conditional distributional invariance"

However, the authors do not run their proposed algorithm on any standard benchmarks, making it difficult to discern whether or not IFM would be useful for practitioners.  Indeed, it looks like IFM isn't even run on the Noised MNIST dataset, which raises concerns about whether IFM on its own works (without the ERM loss term).
* The authors refer to $U_t$ as a "layer" (line 325), which is confusing to me.  Is this supposed to be related to the layers of a DNN?
* It's unclear how the algorithm works in the nonlinear case, i.e. when you need to do feature matching for different layers.  As I mentioned above, a better understanding of how the algorithms are implemented would lead to stronger insights from the experiments.

### Overall evaluation

---

Overall, I liked this paper.  I think that it makes a strong theoretical contribution, and that it studies an important and understudied aspect of DG.  I believe that the paper is generally well written, and I appreciate that although this is more of a theoretical paper, the authors ran some experiments.  This being said, there are a number of drawbacks that I would like to see addressed.  Firstly, the algorithmic description and experimental evaluation are weak compared to the theory, but I think both of these points could be addressed (see above).  Another weakness is the lack of a related work section, which ideally would more broadly discuss recent developments and algorithms proposed for the DG setting.  All of this being said, I'm leaning toward accepting this paper.

### Citations

---

[Zhang et al., 2021] Zhang, Marvin, et al. "Adaptive risk minimization: Learning to adapt to domain shift." Advances in Neural Information Processing Systems 34 (2021): 23664-23678.

[Robey et al., 2021] Robey, Alexander, George J. Pappas, and Hamed Hassani. "Model-based domain generalization." Advances in Neural Information Processing Systems 34 (2021): 20210-20229.

[Gulrajani & Lopez-Paz, 2020] Gulrajani, Ishaan, and David Lopez-Paz. "In search of lost domain generalization." arXiv preprint arXiv:2007.01434 (2020).

[Koh & Sagawa et al., 2021] Koh, Pang Wei, et al. "Wilds: A benchmark of in-the-wild distribution shifts." International Conference on Machine Learning. PMLR, 2021.

---

> ### Author Response · Authors · 2022-08-02
> **Response to Reviewer D1nV**
>
> We thank the reviewer for detailed feedback, and for recognizing the strong theoretical contribution, clear presentation, and importance of the topic.
>
> (1) “Over-simplification of the field of DG in the intro”: We’ve updated the intro line 25-26 to say that feature distribution matching algorithms are **among** the top-performing algorithms on current benchmarks. We think they’re worth studying even though no algorithm is consistently better than ERM across benchmarks. We also added a related works section in Appendix A to broadly discuss literature on DG.
>
> (2) “The term feature-matching is confusing”: ‘Feature-matching’ algorithms enforce some statistics of intermediate representations of examples from different domains to be close. This is usually achieved via a regularizer trained with the supervised loss. CORAL, MMD, DANN as well as our algorithm all follow this paradigm. We added a definition in line 23-25. For our algorithm, as mentioned in line 70-71, IFM matches label-conditioned feature distributions on a small, disjoint subset of the training environments.
>
> (3) “The description of the algorithm that is actually run in practice is confusing”: Thank you for the clarification questions! We’ve updated Alg 1 and line 181-185 in the draft. To respond here as well: the subsets are uniformly randomly drawn from all training envs, without replacement. The algorithm doesn’t require T since we stop once $r_t$ cannot be further reduced. T is a quantity used for analysis. $r_t$ can be found via binary search. For each candidate $r_t$, we perform SGD on objective $\min_{U_t \in \mathbb{R}^{r_t \times r_{t-1}}} \sum_{e, e' \in \mathcal{E}_t}\|U_t (\Sigma^e-\Sigma^{e'}) U_t^T\|_F^2 + \lambda \|U_t^\top U_t - I \|_F^2$. If we can find an $\epsilon$-optimal $U_t$ this means that $r_t$ is admissible.
>
> (4) Why doesn't CORAL work: We do not claim that IFM is theoretically better than CORAL. Our techniques don’t work for CORAL because when matching all training envs at once (equivalent to solving T quadratic equations), there could be a solution that still uses spurious dimensions. The assumption of randomness in covariance matrices and the method of iterative matching are designed to avoid this difficulty. CORAL may also have sublinear environment complexity—we think this is a very interesting question to explore for future work. We conjecture that the reason IFM performs better in some settings (including Gaussian and Noised MNIST) may be that the randomness of batches of training envs used at different layers help the algorithm avoid bad local minima. Though testing this robustly across benchmarks and proving it would certainly be quite non-trivial.
>
> (5) Regarding experiment:
> “​​The Noisy MNIST dataset doesn't seem to satisfy the assumption concerning $Z_2$”: NoisedMNIST indeed has adversarial mean and random covariances for its spurious features, just like our theory. However, since it is a 10-way classification, we do not have $\pm Y$ indicating different classes. We could have different means for different envs, but identical means would make it more challenging for the learner, and hence a strength for our paper rather than a weakness.
> “The authors do not run their proposed algorithm on any standard benchmarks”: Our experiments are not designed to show that IFM is better than CORAL. We never claimed this, nor showed something of that kind theoretically. Rather, our experiments are **inspired** by the theory to see whether some ideas from the analysis could be used to improve existing algorithms. The benefit of IFM is that it comes with theoretical guarantees under our data model — and we’d certainly be happy if someone proved CORAL has comparable performance guarantees.
>
> Nevertheless, we ran some additional experiments on a subset of DomainBed to further demonstrate how our theory can **potentially** inspire algorithmic improvements. We compare CORAL (which matches only the last layer representations) with CORAL_IFM (which also matches representations before the last residual block):

---

> > ### Author Response · Authors · 2022-08-02
> > **Response to Reviewer D1nV (continued)**
> >
> > ========= Dataset: PACS, model selection method: training-domain validation set
> >
> > Algorithm$\quad \quad \quad\quad$A$\quad\quad\quad\quad\quad$C$\quad\quad\quad\quad\quad$P$\quad\quad\quad\quad\quad$S$\quad\quad\quad\quad\quad$Avg
> >
> > CORAL$\quad\quad\quad$82.5 +/- 0.6$\quad$75.8 +/- 0.6$\quad$ 95.1 +/- 0.3$\quad$77.1 +/- 1.0$\quad$82.6 +/- 0.3
> >
> > CORAL_IFM$\quad$81.9 +/- 0.8$\quad$75.7 +/- 0.4$\quad$95.0 +/- 0.2$\quad$78.2 +/- 0.3$\quad$82.7 +/- 0.2
> >
> > ========= Dataset: VLCS, model selection method: training-domain validation set
> >
> > Algorithm$\quad \quad \quad\quad$C$\quad\quad\quad\quad\quad$L$\quad\quad\quad\quad\quad$S$\quad\quad\quad\quad\quad$V                  $\quad\quad\quad\quad\quad$Avg
> >
> > CORAL$\quad\quad\quad$93.7 +/- 1.2$\quad$62.4 +/- 0.7$\quad$70.2 +/- 0.4$\quad$73.2 +/- 0.9$\quad$74.9 +/- 0.4
> >
> > CORAL_IFM$\quad$96.0 +/- 0.3$\quad$63.4 +/- 0.4$\quad$68.9 +/- 0.5$\quad$72.4 +/- 0.9$\quad$75.2 +/- 0.3
> >
> > ========= Dataset: OfficeHome, model selection method: training-domain validation set
> >
> > Algorithm$\quad \quad \quad\quad$A$\quad\quad\quad\quad\quad$C$\quad\quad\quad\quad\quad$P$\quad\quad\quad\quad\quad$R$\quad\quad\quad\quad\quad$Avg
> >
> > CORAL $\quad\quad\quad$56.5 +/- 0.3$\quad$50.4 +/- 0.5$\quad$71.7 +/- 0.2$\quad$73.2 +/- 0.1$\quad$62.9 +/- 0.2
> >
> > CORAL_IFM$\quad$57.6 +/- 0.6$\quad$49.4 +/- 0.6$\quad$71.7 +/- 0.2$\quad$73.3 +/- 0.3$\quad$63.0 +/- 0.2
> >
> > Due to limited time and compute during the author response period, we only ran on the default hyperparameters in DomainBed codebase on ResNet18 for 3K steps (instead of Resnet50 and 5K) over 5 trials. Loss curves show that training already converges at 3K. On all three datasets, CORAL_IFM performs on par with CORAL and even slightly better for VLCS. Although CORAL_IFM does not consistently improve over CORAL, none of the existing algorithms perform significantly better than CORAL anyways (see DomainBed benchmarks). The above shows: (1) One natural extension of IFM to nonlinear setting is to match statistics at multiple layers in a deep NN. (2) At least in some tasks and settings, doing so could improve over baseline. In Section 7 we also experimented with other ideas such as (1) matching different envs at each layer, and (2) regularizing the last dense layer to be orthonormal. Those directions are also extensions to nonlinear models.
> >
> > We hope to reiterate that the main message of the paper doesn’t hinge on those results, since the main message is that one feature matching algorithm **provably** works for some data model, not that iterative matching is necessarily better than non-iterative. If iterative matching turns out better than non-iterative, it is only  “icing on the cake”.
> >
> > (6) “Is $U_t$ supposed to be related to the layers of a DNN?”: Yes. In line 68-69, we drew the connection that iterative matching is similar to matching at multiple layers in a deep neural network.
> >
> > (7) “It’s unclear how the algorithm works in the nonlinear case”: Although IFM is a theoretical algorithm meant only for linear case, one natural extension is conditional feature matching at multiple layers of a deep NN, using a different subset of envs at each layer, and / or regularizing the last feedforward layer to be orthonormal. This is precisely what we already tried in the experiments.

---

> > > ### Author Response · Authors · 2022-08-02
> > > **Response to Reviewer D1nV (continued)**
> > >
> > > Questions:
> > >
> > > (1) Test environment generation: Indeed, some “link” between train environments and test environments is needed. If a completely arbitrary covariance/mean is allowed, there is no such link — however relaxing our assumptions might very well be possible.
> > >
> > > (2) Dimension of style: Indeed, this question seems a bit philosophical in nature, so can be debated. The motivation for our remark was that latent dimensions used for, say, GAN or VAE generative models are usually on the order of hundreds for good reconstruction [1].
> > >
> > > (3) Transversality: Intuitively this condition says several surfaces intersect generically so there is no degeneracy where tangent lines overlap. Concretely, the transversality condition in Theorem 4.2 means that the spurious feature means are linearly independent.
> > >
> > > (4) IID label draw: Thank you, we’ve updated the line after 130.
> > >
> > > (5) Why does $r_t$ shrink: In each iteration IFM discards dimensions where the feature statistics don’t match. If the covariances were completely adversarial (e.g. identical), the spurious dimensions could also match, so $r_t$ could fail to shrink.
> > >
> > > (6) Uniqueness of $w^*$: $w^*$ is unique in the binary classification setting. It is $(\Sigma_1 A^\top)^\dagger \mu_1 $.
> > >
> > > (7) Simple algorithm: We included this algorithm for completeness, but it is indeed orthogonal to our main message and cannot be extended to realistic datasets (it is extremely specific to the linear model and strongly uses Gaussianity), so we didn’t compare it experimentally.
> > >
> > > (8) Class imbalance: Class imbalance induces a bias term in ERM solution so the analysis is more involved, but qualitatively the bound is still constant, i.e. if training accuracy is larger than some threshold than the test accuracy is lower than $\min\{(\eta, 1-\eta)\}$.
> > >
> > > (9) Assumptions for lower bounds: When proving lower bounds, we just need *one setting* of parameters with non-zero measure such ERM/IRM fail, whereas for upper bounds more generality (i.e. more settings of parameters) is a plus. Hence the mismatch isn’t a drawback but a feature of lower / upper bound analysis.
> > >
> > > [1] https://github.com/mafda/generative_adversarial_networks_101

---

> > > > ### Comment · Reviewer_D1nV · 2022-08-07
> > > > **Thanks for your response**
> > > >
> > > > I want to thank the authors for their detailed response.  I think that the added discussion of related work, the updated definitions, and the additional experiments make this paper much stronger.  I also think that the paper will be greatly improved by a more robust algorithm description, so I’m glad to see that that has been added too.
> > > >
> > > > I also agree with the authors here:
> > > >
> > > > > “Our experiments are not designed to show that IFM is better than CORAL. We never claimed this, nor showed something of that kind theoretically. Rather, our experiments are inspired by the theory to see whether some ideas from the analysis could be used to improve existing algorithms.’’
> > > >
> > > > This paper presents a strong theoretical contribution.  While the paper would certainly be stronger if it beat SOTA algorithms on standard benchmarks, this is by no means necessary for acceptance.  The authors have provided more than enough theoretical evidence for their framework.  For this reason, I don’t agree with Reviewer Fzzp here:
> > > >
> > > > > “ From the new experiments, it seems that the IFM + CORAL does not improve the performance of CORAL (or very slightly). Therefore, it remains questionable whether the proposed method can be applied in practice, or it could be that the data model studied is too simple to represent more realistic tasks such as PACS/VLCS/OfficeHome.”
> > > >
> > > > @Fzzp: What are you looking for exactly?  It seems that in order to gain a score higher than a four, this paper would not only need to have a strong theoretical contribution, but it would also need to advance the SOTA algorithmically on standard benchmarks.  I feel that this an unreasonably high expectation.  For instance, if this paper proposed an algorithm that beat SOTA on say WILDS or DomainBed, I would probably upgrade my score to an 8 at least.  But at the end of the day, the question to ask is: Does this paper advance our understanding of DG.  I think yes, and based on your review:
> > > >
> > > > > “ The presentation is clear with solid theoretical results, it involves interesting techniques from probability theory, statistics and differential topology, which might be useful for later theoretical results in this field.”
> > > >
> > > > It seems that you agree that this paper makes a solid contribution.  I would make a similar argument WRT Reviewer ebXq‘s review:
> > > >
> > > > > “The experimental part is relatively weak, several more baselines (refer to Domainbed) and benchmarks (Colored-MNIST or other real-world datasets) should be included for better insight and significance.”
> > > >
> > > > Although that review seems to be below the threshold mandated by the NeurIPS reviewing criteria (as I have already pointed out).
> > > >
> > > > This is all to say that I think this paper makes a solid contribution.  Sure, it doesn’t solve DG once and for all.  But solving DG is not the threshold for being accepted.  In my opinion, the threshold should be whether or not a paper advances our understanding of the field, and I feel strongly that this paper does that.  Therefore, I’m increasing my score.

---

### Official Review · Reviewer_Fzzp · 2022-07-10

**Rating:** 5
**Confidence:** 3
**Soundness:** 2 fair
**Presentation:** 3 good
**Contribution:** 3 good

**Summary:**

This paper analyzes the environmental complexity in domain generalization: how many environments are needed in order to recover an invariant predictor? While most existing papers give pessimistic results on domain generalization algorithms, this work shows positive theoretical results on the sublinear environment complexity. However, the data model may not be practical for real usage.

--post rebuttal--
Thank you for the detailed response and the effort for improving the draft and adding discussions. From the new experiments, it seems that the IFM + CORAL does not improve the performance of CORAL (or very slightly). Therefore, it remains questionable whether the proposed method can be applied in practice, or it could be that the data model studied is too simple to represent more realistic tasks such as PACS/VLCS/OfficeHome. Although the theoretical analysis seems interesting, my main concerns are not addressed. Finally, understanding the non-linear feature embedding is more interesting (though challenging) and I would expect some theoretical results on this case in the revised draft.

--post post rebuttal--
After checking the response of Reviewer D1nV I decide to update my score due to the proof of the IRM lower bound which uses tools from differential topology. I haven't checked the proof in too much detail though.

**Questions:**

See the questions above.

**Ethics Review Area:**

["I don’t know"]

**Limitations:**

As the authors acknowledge in Appendix D, the main limitation is the applicability of IFM in realistic datasets.

**Strengths And Weaknesses:**

Strengths:
1. This work is novel and interesting to the domain generalization field, and analyzes the theoretical model proposed in Rosenfeld et al. The data generation is anti-casual, we first generate the (binary) uniform label distribution and then generate the invariant and spurious features. Finally, the input variable is generated. The novelty part in this work is the addition of a Gram matrix from Gaussian.
2. Based on the data model, the authors propose Iterative Feature Matching, or IFM, for finding invariant featurizer and classifier in $O(\log d_s)$ rounds with $\tilde{O}(1)$ training environments per round. Therefore the environmental complexity is smaller than the $O(d_s)$ complexity of ERM/IRM.
3. The presentation is clear with solid theoretical results, it involves interesting techniques from probability theory, statistics and differential topology, which might be useful for later theoretical results in this field.

Weaknesses:
1. The setting borrows from Rosenfeld et al, which shows the reason why IRM may fail in some simple cases. However, the goal of the current paper is to show an DG algorithm that works in practice. As much as I appreciate the theoretical contributions about the positive results of IFM, in practice (e.g. on MNIST), the algorithm does not work. Moreover, as mentioned in Appendix C, there is even a $O(1)$ environmental complexity algorithm based on Assumption 3.2, which further adds doubts to the applicability of this assumption. If IFM does provide intuitions into algorithmic design, the authors need to at least show some more practical examples.
2. The lower bounds of ERM/IRM do not seem novel compared to Rosenfeld et al. The only difference is the assumption 3.2 vs. assumption 3.1. In Line 149-150, the authors claim that the analysis applies to all but a measure-zero set of covariances, but I'm not quite convinced since the Gram matrix is built from standard normal and the $\bar{\Sigma}$ is bounded.
3. A practical algorithm for finding the maximum dimension $r_t$ in Alg 1 is needed. This is very important for practitioners to apply this algorithm.
4. In the experimental section, the main contribution part is 1) addition of orthonomality of the featurizer in the Gaussian model; 2) matching the sufficient statistics of each layer vs. only matching the last layer. This experiment may be too far away from the linear featurizer and the data model presented in the theoretical analysis. If matching the representation of each layer works for CORAL/DANN/MMD in practice, could the authors try some experiments or prove some theoretical results?
5. In line 203-205, the goal is to provide a theoretical justification of the feature matching algorithm, rather than to solve a specific data model, but the main theoretical results are based on this model.
6. The IFM algorithm resembles Independent Component Analysis (ICA). Discussions and comparisons of the similarities and differences are needed.

---

> ### Author Response · Authors · 2022-08-02
> **Response to Reviewer Fzzp**
>
> We thank the reviewer for recognizing our work for its novelty, clear presentation, and interesting theoretical techniques from probability theory, statistics and differential topology.
>
> (1) “The goal of the current paper is to show an DG algorithm that works in practice” “The algorithm does not work for MNIST”: Our work is a theoretical **analysis** paper rather than an empirical **method** paper. Our goal is to provide one of the first positive **theoretical** results for distribution-matching algorithms under a concrete, natural and non-trivial data model, rather than to propose a new one. As shown in past theoretical works in this field [Rosenfeld et al, Kamath et al, Ahuja et al], proving positive results in even the simplest linear data model is hard for any existing algorithm—and in fact, many popular algorithms like IRM have unreasonable environment complexity. We never claim that IFM is theoretically *better* than other distribution-matching algorithms—and our experiments are not designed to show IFM is better than CORAL or related methods. Our experiments are **inspired** by the theory to see whether some ideas from the analysis could be used to improve existing algorithms. The benefit of IFM is that it comes with theoretical guarantees under our data model — and we’d certainly be happy if someone proved CORAL has comparable performance guarantees.
>
> Nevertheless, we ran some additional experiments on a subset of DomainBed to further demonstrate how our theory can **potentially** inspire algorithmic improvements. We compare CORAL (which matches only the last layer representations) with CORAL_IFM (which also matches representations before the last residual block):
>
> ========= Dataset: PACS, model selection method: training-domain validation set
>
> Algorithm$\quad \quad \quad\quad$A$\quad\quad\quad\quad\quad$C$\quad\quad\quad\quad\quad$P$\quad\quad\quad\quad\quad$S$\quad\quad\quad\quad\quad$Avg
>
> CORAL$\quad\quad\quad$82.5 +/- 0.6$\quad$75.8 +/- 0.6$\quad$ 95.1 +/- 0.3$\quad$77.1 +/- 1.0$\quad$82.6 +/- 0.3
>
> CORAL_IFM$\quad$81.9 +/- 0.8$\quad$75.7 +/- 0.4$\quad$95.0 +/- 0.2$\quad$78.2 +/- 0.3$\quad$82.7 +/- 0.2
>
> ========= Dataset: VLCS, model selection method: training-domain validation set
>
> Algorithm$\quad \quad \quad\quad$C$\quad\quad\quad\quad\quad$L$\quad\quad\quad\quad\quad$S$\quad\quad\quad\quad\quad$V                  $\quad\quad\quad\quad\quad$Avg
>
> CORAL$\quad\quad\quad$93.7 +/- 1.2$\quad$62.4 +/- 0.7$\quad$70.2 +/- 0.4$\quad$73.2 +/- 0.9$\quad$74.9 +/- 0.4
>
> CORAL_IFM$\quad$96.0 +/- 0.3$\quad$63.4 +/- 0.4$\quad$68.9 +/- 0.5$\quad$72.4 +/- 0.9$\quad$75.2 +/- 0.3
>
> ========= Dataset: OfficeHome, model selection method: training-domain validation set
>
> Algorithm$\quad \quad \quad\quad$A$\quad\quad\quad\quad\quad$C$\quad\quad\quad\quad\quad$P$\quad\quad\quad\quad\quad$R$\quad\quad\quad\quad\quad$Avg
>
> CORAL $\quad\quad\quad$56.5 +/- 0.3$\quad$50.4 +/- 0.5$\quad$71.7 +/- 0.2$\quad$73.2 +/- 0.1$\quad$62.9 +/- 0.2
>
> CORAL_IFM$\quad$57.6 +/- 0.6$\quad$49.4 +/- 0.6$\quad$71.7 +/- 0.2$\quad$73.3 +/- 0.3$\quad$63.0 +/- 0.2
>
> Due to limited time and compute during the author response period, we only ran on the default hyperparameters in DomainBed codebase on ResNet18 for 3K steps (instead of Resnet50 and 5K) over 5 trials. Loss curves show that training already converges at 3K. On all three datasets, CORAL_IFM performs on par with CORAL and even slightly better for VLCS. Although CORAL_IFM does not consistently improve over CORAL, none of the existing algorithms perform significantly better than CORAL anyways (see DomainBed benchmarks). The above shows: (1) One natural extension of IFM to nonlinear setting is to match statistics at multiple layers in a deep NN. (2) At least in some tasks and settings, doing so could improve over baseline. In Section 6 we also experimented with other ideas such as (1) matching different envs at each layer, and (2) regularizing the last dense layer to be orthonormal. Those directions are also extensions to nonlinear models.
>
> We hope to reiterate that the main message of the paper doesn’t hinge on those results, since the main message is that one feature matching algorithm **provably** works for some data model, not that iterative matching is necessarily better than non-iterative. If iterative matching turns out better than non-iterative, it is a bonus feature.

---

> > ### Author Response · Authors · 2022-08-02
> > **Response to Reviewer Fzzp (continued)**
> >
> > (2) Regarding the novelty of the lower bounds and the generality of assumption 3.2: Although the ERM lower bound is straightforward, the IRM lower bound is entirely new and requires advanced techniques in differential geometry (see Appendix B.4). One main contribution of our work is showing a separation between featuring matching algorithms and ERM / IRM in a concrete data model. The fact that we can establish such a separation with a simple and subtle modification to an existing model is a strength, rather than a weakness. Regarding Assumption 3.2: the assumption says the covariance can be adversarially chosen *as long as there is a small Gaussian noise* added. Our bound on environment complexity depends only logarithmically on the norm bound so the requirement is mild. Theoretically a norm bound is necessary, since when $\overline{\Sigma_2}$ is large, a small noise is almost undetectable. We’ve updated Line 150-151 for clarity.
> >
> > (3) How to find $r_t$ in practice: We can find $r_t$ using binary search between 1 and $r_{t-1}$. Namely, for a “guess” of $r_t$, we perform SGD from random initialization to train $U_t$ to minimize average $\|U (\Sigma_e-\Sigma_{e'}) U^T \|_F^2$ over pairs of training env $e, e'$, plus a regularization term $| U^T U - I |_F^2$
> >
> > to enforce orthonormality. In our experiments we simply halve the dimensions $r_t = r_{t-1}/2$ which works well. We appreciate this question, and for clarity *we have updated Alg 1 in our draft with more detailed algorithmic descriptions.*
> >
> > (4) “Could the authors try some experiments or prove some theoretical results [in nonlinear setting]?”: For additional experiments on matching multiple layers, refer to (1). Extending our theory to nonlinear settings is an important (and challenging) future direction.
> >
> > (5) “The main theoretical results are based on [a specific data] model.”: Indeed domain generalization is a very challenging problem, and absent assumptions on the data distributions (i.e. what constitutes a valid new domain/environment) it’s clearly ill-defined — so any positive result is contingent upon *some* assumptions. While our data model is simple, it has already been used as a sandbox to understand algorithms like IRM [Rosenfeld et al. Kamath et al.] because it captures some important aspects of real-life data like latent variables and correlations between the labels and the spurious features.
> >
> > (6) Discussions and comparisons of IFM with ICA: We added a comparison with ICA to Appendix A in our draft. Our data model bears some semblance to ICA since the invariant and spurious features are independent and the goal is to “disentangle”/”separate” the two sets of variables/sources. However, for identifiability in linear ICA at most one source has to be non-Gaussian, which is different from our data model. IFM is also distinct from ICA methods such as minimization of mutual information or maximization of non-Gaussianity. A line of recent work on nonlinear ICA [Hyvarinen et al. 16', 19', Khemakhem et al. 20'] proves identifiability results for deep latent variable models but they require additional conditions or auxiliary supervision. Ultimately, we think of these lines of work as providing orthogonal approaches, and understanding when they are a better (or worse choice) than IFM (or really, any competitor methods for domain generalization) is an important direction for further work.

---

### Official Review · Reviewer_ebXq · 2022-07-12

**Rating:** 4
**Confidence:** 4
**Soundness:** 3 good
**Presentation:** 2 fair
**Contribution:** 3 good

**Summary:**

This paper proposes an iterative feature matching algorithm to solve domain generalization problem with logarithmic environment complexity, compared to linear complexity of ERM and IRM. The authors also provide theoretical justification and empirical evaluation.

**Questions:**

Refer to Weakness

**Limitations:**

Refer to Weakness

**Strengths And Weaknesses:**

Strength:
(a)	The investigated problem is of great importance in invariance-based domain generalization and out-of-distribution generalization.
(b)	The proposed method has sound theoretical justification.
Weakness:
(a)	The experimental part is relatively weak, several more baselines (refer to Domainbed) and benchmarks (Colored-MNIST or other real-world datasets) should be included for better insight and significance.
(b)	The organization of the manuscript could be improved by moving proofs into appendix rather than piling them up in main body.
(c)	The authors should elaborate more on the gap between linear setting in theoretical analysis and real world applications with more complex models.

---

> ### Author Response · Authors · 2022-08-02
> **Response to Reviewer ebXq**
>
> We thank the reviewer for recognizing the importance of the topic and the solidity of our theoretical contributions.
>
> (1) **Regarding weak experimentation**: The main goal of our paper is to provide one of the first positive **theoretical** results for distribution-matching algorithms under a concrete, natural and non-trivial data model, rather than to propose a new method. As shown in past theoretical works in this field [Rosenfeld et al, Kamath et al, Ahuja et al], proving positive results in even the simplest linear data model is hard for any existing algorithm—and in fact, many popular algorithms like IRM have unreasonable environment complexity. We never claim that IFM is theoretically *better* than other distribution-matching algorithms—and our experiments are not designed to show IFM is better than CORAL or related methods. Our experiments are **inspired** by the theory to see whether some ideas from the analysis could be used to improve existing algorithms. The benefit of IFM is that it comes with theoretical guarantees under our data model — and we’d certainly be happy if someone proved CORAL has comparable performance guarantees.
>
> Nevertheless, we ran some additional experiments on a subset of DomainBed to further demonstrate how our theory can **potentially** inspire algorithmic improvements. We compare CORAL (which matches only the last layer representations) with CORAL_IFM (which also matches representations before the last residual block):
>
> ========= Dataset: PACS, model selection method: training-domain validation set
>
> Algorithm$\quad \quad \quad\quad$A$\quad\quad\quad\quad\quad$C$\quad\quad\quad\quad\quad$P$\quad\quad\quad\quad\quad$S$\quad\quad\quad\quad\quad$Avg
>
> CORAL$\quad\quad\quad$82.5 +/- 0.6$\quad$75.8 +/- 0.6$\quad$ 95.1 +/- 0.3$\quad$77.1 +/- 1.0$\quad$82.6 +/- 0.3
>
> CORAL_IFM$\quad$81.9 +/- 0.8$\quad$75.7 +/- 0.4$\quad$95.0 +/- 0.2$\quad$78.2 +/- 0.3$\quad$82.7 +/- 0.2
>
> ========= Dataset: VLCS, model selection method: training-domain validation set
>
> Algorithm$\quad \quad \quad\quad$C$\quad\quad\quad\quad\quad$L$\quad\quad\quad\quad\quad$S$\quad\quad\quad\quad\quad$V                  $\quad\quad\quad\quad\quad$Avg
>
> CORAL$\quad\quad\quad$93.7 +/- 1.2$\quad$62.4 +/- 0.7$\quad$70.2 +/- 0.4$\quad$73.2 +/- 0.9$\quad$74.9 +/- 0.4
>
> CORAL_IFM$\quad$96.0 +/- 0.3$\quad$63.4 +/- 0.4$\quad$68.9 +/- 0.5$\quad$72.4 +/- 0.9$\quad$75.2 +/- 0.3
>
> ========= Dataset: OfficeHome, model selection method: training-domain validation set
>
> Algorithm$\quad \quad \quad\quad$A$\quad\quad\quad\quad\quad$C$\quad\quad\quad\quad\quad$P$\quad\quad\quad\quad\quad$R$\quad\quad\quad\quad\quad$Avg
>
> CORAL $\quad\quad\quad$56.5 +/- 0.3$\quad$50.4 +/- 0.5$\quad$71.7 +/- 0.2$\quad$73.2 +/- 0.1$\quad$62.9 +/- 0.2
>
> CORAL_IFM$\quad$57.6 +/- 0.6$\quad$49.4 +/- 0.6$\quad$71.7 +/- 0.2$\quad$73.3 +/- 0.3$\quad$63.0 +/- 0.2
>
> Due to limited time and compute during the author response period, we only ran on the default hyperparameters in DomainBed codebase on ResNet18 for 3K steps (instead of Resnet50 and 5K) over 5 trials. Loss curves show that training already converges at 3K. On all three datasets, CORAL_IFM performs on par with CORAL and even slightly better for VLCS. Although CORAL_IFM does not consistently improve over CORAL, none of the existing algorithms perform significantly better than CORAL anyways (see DomainBed benchmarks). The above shows: (1) One natural extension of IFM to nonlinear setting is to match statistics at multiple layers in a deep NN. (2) At least in some tasks and settings, doing so could improve over baseline. In Section 7 we also experimented with other ideas such as (1) matching different envs at each layer, and (2) regularizing the last dense layer to be orthonormal. Those directions are also extensions to nonlinear models.
>
> We hope to reiterate that the main message of the paper doesn’t hinge on those results, since the main message is that one feature matching algorithm **provably** works for some data model, not that iterative matching is necessarily better than non-iterative. If iterative matching turns out better than non-iterative, it is a bonus feature.
>
> (2) We moved the proof sketch of Lemma 5.2 to the Appendix B.1 in the updated draft. Thanks for the suggestion!
>
> (3) As mentioned in (1), theoretical analysis in even the simplest linear setting is non-trivial and we leave analysis in nonlinear settings to future work. Empirically, we already demonstrated in Section 7 and in the additional experiments above that for practitioners, our theory offers ideas for matching at more layers and regularization in a deep NN.

---

> > ### Comment · Reviewer_ebXq · 2022-08-09
> > **Some left questions about Colored-MNIST**
> >
> > Thank the authors for the response, it did addressed several concerns, and I have some further questions. As for the proposed problem setting, I think the Colored MNIST [IRM] perfectly fits this setting and is a simple experiment which is easy to run. From the results of IRM, I find that with only 2 environments, it is enough to achieve a good OOD generalization performance. However, the theoretical analysis in this work establish a lower bound of IRM, which proves the number of environments is at least O(d_s). I think there may be some mismatch here (or in this setting d_s=1?). Therefore, I really want to see the results of the Colored MNSIT experiment.
> > Secondly, in the newly-added experiments, I wonder the scale of d_s in this setting. How large is d_s? I think it is hard to analyze this.
> > I would re-evaluate the score if I could see some analysis.

---

> > > ### Author Response · Authors · 2022-08-09
> > > **Re: Some left questions about Colored-MNIST**
> > >
> > > We thank the reviewer for joining the discussion and hope the following answers address your lingering concerns:
> > >
> > > (1) “From the results of IRM, I find that with only 2 environments, it is enough to achieve a good OOD generalization performance.”:
> > >
> > > We would like to point out that the values reported in the original IRM paper are **invalid**: they are the result of hyperparameter tuning on the test set. This is acknowledged in their code release. For a fair evaluation, see [Gulrajani and Lopez-Paz], who showed that IRM actually does not have significantly better performance than ERM or CORAL (both around 0.52 avg accuracy).
> > >
> > > (2) “The Colored MNIST [IRM] perfectly fits this setting” “I think there may be some mismatch here (or in this setting d_s=1?)”
> > >
> > > We’re not quite sure what setting the “this” is referring to. If you mean our data model, no “real life” dataset matches our model exactly — the Gaussian dataset in Figure 1 is directly generated from our model so the closest match to it. Noised MNIST seems a closer match to the scenario we are interested in than Colored MNIST, since the noise added is high-dimensional.
> > >
> > > (3) “Therefore, I really want to see the results of ColoredMNIST”:
> > >
> > > We again stress that we chose Noised MNIST as a closer match to the difficulty that our theory and algorithm are trying to ameliorate: high-dimensional “spurious” features — since the noise added is high-dimensional.  So we expect the gap between the performance of IFM and IRM to be a lot bigger on Noised MNIST. Nevertheless, we ran experiments on ColoredMNIST to show that empirically IFM has comparable performance even on a dataset that was specifically designed for IRM in [Arjovsky et al.]:
> > >
> > > === Dataset: ColoredMNIST, model selection method: training-domain validation set
> > >
> > > Algorithm$\qquad$+90%$\qquad\qquad$+80%$\qquad\qquad$-90%$\qquad\qquad$Avg
> > >
> > > CORAL_IFM$\qquad$71.7 +/- 0.2$\qquad$72.9 +/- 0.1$\qquad$10.1 +/- 0.1$\qquad$51.6
> > >
> > > In comparison, DomainBed reports 51.5 +/- 0.2 for CORAL and 52.0 +/-0.1 for IRM. Therefore CORAL_IFM is at least no worse than IRM even on a dataset that was **specifically designed for IRM**.
> > >
> > > (4) “How large is $d_s$ in the newly added experiments?”
> > >
> > > A real-life dataset (i.e. one that isn’t specifically generated from a data model) doesn’t have “ground truth features” —  so there isn’t a way to definitively calculate $d_s$. In NoisedMNIST the spurious noise is high-dimensional, so we would expect it to behave similarly as our model for $d_s$ that is high.
> > >
> > > More generally, it is reasonable to model stylistic variations as high-dimensional in realistic DG tasks like VLCS, PACS, and Office_Home. As we wrote in our response to Reviewer D1nV regarding ‘Dimension of style’, one piece of evidence for this is that the latent dimensions used in GAN and VAE for style transfer are usually on the order of hundreds (e.g. 256 for ST-VAE [Liu et al]). Therefore, an environment complexity of $O(d_s)$ is prohibitive for realistic datasets.
> > >
> > > We sincerely hope Reviewer ebXq can reevaluate our score if the above answers make sense. Thanks!
> > >
> > >
> > > [Arjovsky et al] "Invariant risk minimization." arXiv preprint arXiv:1907.02893 (2019).
> > >
> > > [Gulrajani and Lopez-Paz] "In search of lost domain generalization." arXiv preprint arXiv:2007.01434 (2020).
> > >
> > > [Liu et al] "Multiple style transfer via variational autoencoder." 2021 IEEE International Conference on Image Processing (ICIP). IEEE, 2021. https://github.com/Holmes-Alan/ST-VAE/blob/main/eval_multiple_style.py

---

> ### Comment · Reviewer_D1nV · 2022-08-03
> **Regarding this review**
>
> I'm also serving as a reviewer on this paper.  I would like to ask Reviewer ebXq to write a more detailed review.  The current review gives **very** little feedback to the authors while advocating for rejection.  If the reviewer is to advocate for rejection, it is their responsibility to give an adequate and detailed reason for this; see the reviewing guidelines (https://nips.cc/Conferences/2020/PaperInformation/ReviewerGuidelines), which state that:
>
> * It is not fair to dismiss any submission without having thoroughly read it. Think about the times when you received an unfair, unjustified, short, or dismissive review. Try not to be that reviewer! Always be constructive and help the authors understand your viewpoint, without being dismissive or using inappropriate language. If you need to cite existing work to justify one of your comments, please be as precise as possible and give a complete citation.
>
> * If you would like the authors to clarify something during the author response phase, please articulate this clearly in your review (e.g., “I would like to see results of experiment X” or “Can you please include details about the parameter settings used for experiment Y”).
>
> For instance, with regard to your weaknesses:
>
> * (a):  What "insight" and/or "significance" are you looking to see?  Why do you think it's important for the authors to include more datasets?
> * (b) Which proofs should be moved to the appendix?  Which results would be made more clear by this?
> * (c) What would be the impact of such an elaboration?  Why was the initial presentation confusing to you?  How **specifically** could the authors improve the paper?
>
> To summarize, it's difficult as a reviewer to engage with this review, because it doesn't demonstrate that the reviewer has thought
>  deeply about this paper and/or read it closely.  And I'd imagine that as an author, receiving such a review is frustrating for similar reasons.  As a community, I think we should aspire to write reviews similar in quality to the reviews that we hope to receive.  And I feel that this review falls well short of this standard.

---

> ### Author Response · Authors · 2022-08-08
> **Reminder for Reviewer ebXq**
>
> Dear Reviewer ebXq,
>
> Thank you for putting in the time and effort to review our paper. As Reviewer D1nV pointed out, our theoretical contributions indeed contribute to better understanding of DG. We have updated our experiments, moved part of the proof sketch to appendix, and elaborated on the main message of the theoretical results, as well as on how the theory can be used to design algorithms applicable to more realistic datasets. We'd love to hear back. If we've sufficiently addressed your concern, could you please reevaluate our score?
>
> Thank you very much!

---

### Comment · Area_Chair_iaDa · 2022-08-09
**Follow up questions for the authors?**

Hi Reviewers, please read the responses by the authors and the other reviews, and ask any follow-up questions/clarifications to the authors as soon as possible. Note that the author-reviewer interaction period ends within a day.

And thanks to the authors for your responses.

Best, The AC for "Iterative Feature Matching: Toward Provable Domain Generalization with Logarithmic Environments"

---

### Meta-Review · Area_Chair_iaDa · 2022-08-30

**Recommendation:** Accept
**Confidence:** Less certain

**Metareview:**

This paper gives theoretical guarantees for the problem of domain generalization from a few training environments. This paper gives an algorithm that achieves strong theoretical improvements over prior work (logarithmically many training environments vs linear number for existing methods).  The reviewers also felt that there are novel insights that may be of independent interest in this space.  There was some concern by some reviewers about the extent of the empirical evaluations. However, since this is primarily a theoretical paper with strong improvements, the contributions seemed above the bar for NeurIPS.

**Award:**

No

---

### Decision · Program_Chairs · 2022-09-14

Accept